# High Prevalence of Alterations in DNA Mismatch Repair Genes of Lynch Syndrome in Pediatric Patients with Adrenocortical Tumors Carrying a Germline Mutation on *TP53*

**DOI:** 10.3390/cancers12030621

**Published:** 2020-03-07

**Authors:** Vania Balderrama Brondani, Luciana Montenegro, Amanda Meneses Ferreira Lacombe, Breno Marchiori Magalhães, Mirian Yumie Nishi, Mariana Ferreira de Assis Funari, Amanda de Moraes Narcizo, Lais Cavalca Cardoso, Sheila Aparecida Coelho Siqueira, Maria Claudia Nogueira Zerbini, Francisco Tibor Denes, Ana Claudia Latronico, Berenice Bilharinho Mendonca, Madson Queiroz Almeida, Antonio Marcondes Lerario, Ibere Cauduro Soares, Maria Candida Barisson Villares Fragoso

**Affiliations:** 1Laboratório de Hormônios e Genética Molecular LIM/42, Unidade de Suprarrenal, Serviço de Endocrinologia e Metabologia, Hospital das Clínicas, Faculdade de Medicina da Universidade de São Paulo, São Paulo 0540396, Brasil; amflacombe@usp.br (A.M.F.L.); breno.marchiori@fm.usp.br (B.M.M.); anacl@usp.br (A.C.L.); beremen@usp.br (B.B.M.); madson.a@hc.fm.usp.br (M.Q.A.); 2Laboratório de Hormônios e Genética Molecular LIM/42, Serviço de Endocrinologia e Metabologia, Hospital das Clínicas, Faculdade de Medicina da Universidade de São Paulo, São Paulo 0540396, Brasilmarianafunari@usp.br (M.F.d.A.F.); 3Laboratório de Sequenciamento em Larga Escala (SELA), Faculdade de Medicina da Universidade de São Paulo, São Paulo 0540396, Brasil; amnarcizo@usp.br (A.d.M.N.); lais.cavalca@fm.usp.br (L.C.C.); 4Departamento de Anatomia Patológica, Faculdade de Medicina da Universidade de São Paulo, São Paulo 0540396, Brasil; sheila.siqueira@hc.fm.usp.br (S.A.C.S.); czerbini@usp.br (M.C.N.Z.); 5Serviço de Urologia, Hospital das Clínicas, Faculdade de Medicina da Universidade de São Paulo, São Paulo 0540396, Brasil; francisco.denes@hc.fm.usp.br; 6Serviço de Endocrinologia, Instituto do Câncer do Estado de São Paulo (ICESP), Faculdade de Medicina da Universidade de São Paulo, São Paulo 0540396, Brasil; 7Department of Internal Medicine, Division of Metabolism, Endocrinology and Diabetes, University of Michigan, Ann Arbor, MI 48109, USA; alerario@umich.edu; 8Serviço de Anatomia Patológica, Instituto do Câncer do Estado de São Paulo (ICESP), Faculdade de Medicina da Universidade de São Paulo, São Paulo 0540396, Brasil; ibere.soares@hc.fm.usp

**Keywords:** adrenal tumor, DNA mismatch repair, *TP53*

## Abstract

Adrenocortical cancer is a rare malignant neoplasm associated with a dismal prognosis. Identification of the molecular pathways involved in adrenal tumorigenesis is essential for a better understanding of the disease mechanism and improvement of its treatment. The aim of this study is to define the prevalence of alterations in DNA mismatch repair (MMR) genes in Lynch syndrome among pediatric patients with adrenocortical neoplasia from southern Brazil, where the prevalence of a specific *TP53* germline mutation (p.Arg337His) is quite high. Thirty-six pediatric patients were retrospectively evaluated. Immunohistochemistry (IHC) for the MMR enzymes MLH1, MSH2, MSH6, and PMS2, as well as next-generation sequencing (NGS) were performed. For IHC, 36 pediatric tumors were tested. In all of them, the expression of all evaluated MMR proteins was well-preserved. For NGS, 35 patients with pediatric tumor were tested. Three patients (8.57%) with the *TP53* p.Arg337His germline mutation presented pathogenic and likely pathogenic variants in the *MMR* genes (two in *MLH1* and one in *MSH6*). The prevalence of altered *MMR* genes among pediatric patients was elevated (8.57%) and higher than in colorectal and endometrial cancer cohorts. Pediatric patients with adrenocortical tumors should, thus, be strongly considered as at genetic risk for Lynch syndrome.

## 1. Introduction

Adrenocortical cancer (ACC) is a rare malignant neoplasm, presenting a general incidence of one to two cases/million/year [1,2] and an overall age-adjusted incidence of 0.72 cases/million/year in the United States [3]. ACC has been associated with a variable clinical presentation [4]; in most cases, it is characterized by hormonal hypersecretion [5], and Cushing’s syndrome is responsible for one-third of cases [3]. ACC usually presents a dismal prognosis [3]. On the other hand, pediatric patients with adrenocortical tumors (ACTs) generally have better prognoses than adult patients [6,7,8]. The incidence of pediatric adrenocortical tumors in Southern Brazil has been estimated to be 3.4–4.2 cases/million children/year [6], which represents a rate that is ten to fifteen times higher than the global incidence rate. This notorious difference is due to the high prevalence of the germline mutation *TP53* p.Arg337His in this region of Brazil [6,9,10,11,12]. Among pediatric patients, germline mutations are responsible for most cases and *TP53* mutations are present in 50–90% of cases [5,11]; this differs from the adult ACC group, in which sporadic cases are the most common [5]. Li-Fraumeni syndrome (LFS), first described in 1969, is an autosomal dominant cancer predisposition syndrome in pediatric cases and in young adults, which results from germline mutations in the *TP53* tumor suppressor gene [13]. LFS families present ACC in 3–10% of cases, showing that germline mutations in *TP53* are linked to adrenal tumorigenesis [14]. Other genetic syndromes have been related to the occurrence of ACT and ACC, such as Beckwith–Wiedemann syndrome (BWS), multiple endocrine neoplasia type 1 (MEN1), familial adenomatous polyposis (FAP), neurofibromatosis type 1, Werner syndrome, and Lynch syndrome (LS) [5]. The identification of molecular pathways responsible for adrenal tumorigenesis is essential for a better understanding of the disease and better treatments for the patient.

LS is an autosomal dominant inherited condition. It was first described in 1966 by Lynch et al. [15] in two large families from Nebraska and Michigan. Although the Nebraska proband presented ACC, adrenal tumors are not recognized as LS-associated tumors and are not mentioned in the clinical criteria for diagnosis. Amsterdam I and II clinical criteria [16] and the Revised Bethesda Guidelines [17] are the clinical criteria used for LS diagnosis, which are based on the occurrence of colorectal cancer (CCR) and the presence of familial cancer history. They also indicate LS-associated tumors, such as endometrial, small bowel, ureter, and pelvis renal cancers. Germline mutations in *MMR* genes (*MLH1, MSH2, MSH6*, and *PMS2*) have been associated with this syndrome since 1993 [18,19,20,21]. Germline mutations in *MLH1*, *MSH2,* and *MSH6* account for more than 90% of cases [22]. Inactivation of one of the *MMR* genes causes loss of MMR protein expression and is responsible for LS [22,23]. Other genetic alterations can cause LS-like disease with a clinical presentation similar to LS, such as the germline mutation in the *EPCAM* gene that causes *MSH2* silencing and the somatic mutation on *BRAF* gene that causes *MLH1* methylation [22,24]. In 2013, Raymond et al. [25] showed, in an adult cohort of ACC, an important prevalence of LS of 3.2%, which is comparable to the prevalence of colorectal and endometrial cancer in LS patients. This important contribution suggests the role of *MMR* genes in adrenal tumorigenesis. 

The aim of this study is to define the prevalence of alterations in *MMR* genes of LS among pediatric patients with adrenocortical tumor.

## 2. Results

### 2.1. Patients

Thirty-six pediatric patients with adrenocortical tumor (ACT) were included in this retrospective study. Twenty-eight patients had their tumor tissue studied by IHC/Tissue microarray (TMA) and eight cases were studied by IHC/Formalin-fixed and paraffin-embedded tumor sections (FFPE). All of them underwent genetic analysis. Twenty-three (63.9%) were female, with a F:M ratio of 1.76:1. The median age was 2.2 years (ranging from 0.8 to 15 years old). Nineteen (55.88%) patients presented isolated virilization, fourteen (41.17%) presented virilization associated with hypercortisolism, and one (2.94%) presented isolated hypercortisolism (two patients were excluded from this analysis, due to a lack of data). All patients underwent the *TP53* genetic test, from which it was found that 33 (91.6%) patients were positive for *TP53* germline pathogenic variants. Thirty-one (31/33, 93.9%) patients exhibited the classical c.1010G > A variant on exon 10 (p.Arg337His), while only two patients presented others described germline pathogenic variants: patient 24, c.818G > A (p.Arg273His); and patient 26, c.733G > A (p.Gly245Ser). All patients underwent surgical resection of the adrenocortical tumors, and histopathology examination was performed by an expert pathologist (MCNZ) and the adrenocortical origin of the tumors confirmed. This cohort presented ACTs with a median weight of 38.95 g (3–1230 g) and a median size of 5 cm (1.5–20 cm). Six patients (16.6%) showed metastatic disease and 3 patients had a fatal outcome. The median follow-up time was 8.6 years (0.03–31.6 years; see Appendix A).

### 2.2. Immunohistochemistry

Of the 45 pediatric ACTs assembled on a previous TMA, nine were excluded due to technical issues in the staining procedure (no internal positive control). Of the remaining 36 tumors, only 28 cases presented available DNA for genetic analysis, who were included in this study. Eight pediatric patients with ACT and with DNA available for genetic analysis were posteriorly included, and their tumor tissue underwent FFPE for IHC.

All the cases studied (i.e., with TMA and with FFPE) presented preserved MLH1/PMS2 and MSH2/MSH6 immunoexpression (positive nuclear staining for all four mismatch repair enzymes; Figure 1).

### 2.3. Genetic Analysis

Thirty-six patients underwent next-generation sequencing (NGS). One patient (patient 29) was excluded from this genetic analysis, due to low-quality DNA. Six patients (17.1%) presented alterations in *MMR* genes (Table 1; Appendix A).

#### 2.3.1. Pathogenic and Likely Pathogenic Variants

Three of thirty-five (8.57%) patients presented germline pathogenic and likely pathogenic variants on *MMR* genes identified on NGS, and all of them presented the *TP53* germline mutation (c.1010G > A, p.Arg337His). The expression of MMR proteins was preserved in immunohistochemistry analyses. Details of NGS are described below.

**Patient #6**. NGS analyses revealed an in-frame deletion variant in the *MLH1* gene [c.1500_1502del, p.Ile501del (rs587778920)]. This variant was localized in a hot spot area with 23 described non-VUSs. gnomAD exome data showed a low allele frequency (0.0000119), and the variant was not found in the gnomAD genome data. It is an in-frame deletion in a non-repeat area and computational prediction analysis also classified it as a pathogenic variant. This variant was classified as class 3/uncertain in the InSiGHT variant database, presenting a probability of pathogenicity between 0.05–0.949 using a multifactorial likelihood model.

**Patient #9**. NGS analyses identified a missense variant in the *MLH1* gene [c.1808C > T, p.Pro603Leu (rs63750876)]. This is a rare variant with a gnomAD exome allele frequency of 0.0000159. It is localized in a hot spot area, with 25 described non-VUSs in the area, and 11 prediction programs classified it as a pathogenic variant (DANN, DEOGEN2, EIGEN, FATHMM-MKL, M-CAP, MVP, MutationAssessor, MutationTaster, PrimateAI, REVEL, and SIFT).

**Patient #33**. NGS analyses showed a nonsense variant in the *MSH6* gene [c.328C > T, p.Arg110* (rs63750019)] (Figure 2). Nonsense variants in *MSH6* are a known mechanism of disease. This specific variant is rare and is localized in a hot spot area; computational analysis programs predicted the variant as pathogenic, and ClinVar lists it as pathogenic.

#### 2.3.2. Variant of Uncertain Significance (VUS)

Three out of thirty-five (8.57%) patients presented a variant of uncertain significance (VUS) in NGS, and all of them presented the *TP53* germline mutation (c.1010G > A, p.Arg337His) and preserved protein expression of MMR proteins in immunohistochemistry analyses.

**Patient #16**. Patient 16 was a female patient of 3.3 years who presented virilization associated with hypercortisolism at clinical admission. During the surgical procedure, the tumor capsule ruptured. After six months, disease recurrence was diagnosed and chemotherapy (etoposide, doxorubicin, and cisplatin plus mitotane) was prescribed. Although chemotherapy was administered, there was disease progression and a new surgical procedure was indicated (partial hepatectomy). Another chemotherapy cycle was prescribed, resulting in stabilized disease. The mitotane was withdrawn due to side effects and the patient is under recurrence surveillance. She has been followed for 8.07 years.

NGS data showed a missense variant in *MSH6* (c.2420A > G, p.Tyr807Cys) that has not been previously described as being associated with Lynch syndrome. However, it is a rare variant not present in the gnomAD bank (exome and genome), and computational predictions from DANN, DEOGEN2, EIGEN, FATHMM-MKL, M-CAP, MVP, MutationTaster, PrimateAI, REVEL, and SIFT ranked it as pathogenic; while MutationAssessor predicted it as benign.

**Patient #21**. NGS analysis found a missense variant in *MSH6* [c.359T > C, p.Ile120Thr (rs775971872)]. This variant is rare (gnomAD exome allele frequency less than 0.001) and is located in a mutational hot spot area with 6 described pathogenic variants. This variant has already been published and associated to LS [26,27].

**Patient #35**. Although there was no disease recurrence or metastasis, the patient had a fatal outcome due to surgical complications. NGS showed a missense variant in *MSH6* [c.643G > T, p.Val215Leu (rs145959653)]. In addition, this variant is not in the gnomAD exome or genome data sets, nine computational prediction sites considered it as benign (DANN, DEOGEN2, EIGEN, FATHMM-MKL, MutationAssessor, MutationTaster, PrimateAI, REVEL, and SIFT), and only one site predicted it to be pathogenic (M-CAP). Additionally, the position was not conserved. This variant was considered as likely benign in the InSight variant database, although it was not classified in the MMR gene variant classification criteria. This germline variant has been reported in a patient with endometrial tumor; unfortunately, no detail of clinical evolution nor outcome was published [28].

#### 2.3.3. Copy Number Variation

MLPA was performed to confirm the results of copy number variation given by CONTRA. MLPA specific to *MLH1/MSH2* showed normal copy numbers of these genes.

### 2.4. Family History of Lynch Syndrome-Associated Tumors

Five patients (#6, #9, #16, #21, and #33) were investigated in terms of their family history of Lynch syndrome-associated tumors, four of which presented family histories of tumors (#6: skin, breast, and mouth cancer, and leukemia; #16: breast and colorectal cancer; #21: breast cancer and adrenocortical carcinoma; and #33: prostate and breast cancer). However, only two patients (#6 and #16) had family histories with tumors associated to Lynch syndrome (skin cancer and colorectal carcinoma, respectively). At present, patient #9 did not identify any cases of tumors in their family. Unfortunately, we had no data regarding the family history of patient #35.

## 3. Discussion

DNA mismatch repair is a system which guarantees the post-replicative integrity of the genome [23]. Heterozygous mutations in one of the four genes involved in *MMR* (*MLH1, MSH2, MSH6,* and *PMS2*) are the cause of Lynch syndrome, while biallelic mutations outline a more severe phenotype and are used to identify constitutional mismatch repair deficiency syndrome [23,29]. The MMR system detects and corrects spontaneous errors that occur during DNA replication [30]. If the function of the system is impaired, genes that regulate cell division can become inactivated, leading to failure to control cell growth and differentiation, which consequently stimulates tumorigenesis [31]. Germline mutations in *MMR* genes cause loss of function of the encoded protein, which results in microsatellite instability (MSI) and, thus, facilitates a hypermutated status [30]. The loss of MMR protein expression has led to the use of immunohistochemistry (IHC) to determine the levels of MMR proteins as markers for deeper issues. Aberrant MMR protein levels, similar to the presence of MSI, are considered as a sign that the MMR system is not functioning normally [23,30]. Although MSI and IHC are used as the first steps in LS screening, their clinical sensitivities are 69–85% and 83%, and clinical specificities are 90.2% and 88.8%, respectively [32].

As LS is a cancer predisposition syndrome, patients with LS present a lifetime risk of colorectal cancer between 30–70%, which is 5–12-fold higher than that in the general population [16,23,33,34]. Regarding endometrial cancer, the cumulative risk of LS patients by 70 years is between 32% and 42%; this risk is 12–16-fold higher than that found in the general female population [22,31,35]. Patients with LS are at an increased risk for the developing colorectal and extracolonic cancers at an early age [22]. Although LS is rare, it is associated with significant morbidity and mortality [36].

This pediatric cohort with ACT was subjected to NGS and a complimentary workup to define the prevalence of alterations in *MMR* genes. This tested cohort was of particular interest, due to the high prevalence of *TP53* mutations, mainly the c.1010G > A, p.Arg337His mutation. Unfortunately, we could not apply the Amsterdam and Bethesda clinical criteria before genetic analyses, although we conducted familial genetic segregation in first-degree relatives. Pathogenic variants on the *MMR* genes were found in 8.57% (3/35) of our pediatric cohort (two were localized in *MLH1,* and one in *MSH6*) and VUS were found in 8.57% (3/35). All VUS were localized in the *MSH6* gene. Interestingly, 3464 variants have been described in *MSH6,* according to UniProt, ClinVar, Varsome, and Pubmed, 1916 of which are missense variants. In these missense variants, 1844 (96.24%) were classified as VUS, the frequency of which is considered high, when compared to other genes such as *TP53* and *CTNNB1*, which showed 66.98% and 22.72% missense variants as VUS, respectively. The International Society for Gastrointestinal Hereditary Tumors (InSiGHT) [37] has suggested a new scheme for the standardized classification of MMR variants, categorizing the variants into five classes of pathogenicity. Our six variants were classified with this InSiGHT scheme and only #6 (*MLH1,* c.1500_1502del, p.Ile501del) was found, which was categorized as Class 3/Uncertain. Class 3 variants present a probability of pathogenicity between 0.05–0.949, according to the InSiGHT multifactorial likelihood model. The #9, #16, #21, #33, and #35 variants were not registered in the InSiGHT *MMR* classification. Dominguez-Valentin et al. [38] reported the findings of the Prospective Lynch Syndrome Database, classifying the variants using the InSiGHT scheme, and proposed that LS should be considered as a generic term for a group of four clinically different inherited cancer risk syndromes. This new suggestion is due to the different results of cancer risk by gene, age, and gender of the class 4 and 5 *MMR* variants [38].

When we compared our data with that of a previous study with adult cohorts associated with LS-related cancers, we noticed an important prevalence of *MMR* variants in our ACT pediatric cohort. Adult colorectal cancer (CRC) cohorts show a prevalence of LS of approximately 3% [39,40], endometrial cancer cohorts demonstrate a LS prevalence of 2.1% [41], and ACC adult cohorts present a prevalence of 3.2% [25].

One point that should considered is related to IHC. We presented three patients with *MMR* pathogenic variants that probably alter the encoded protein; however, the IHC analyses showed positive immunoexpression of all MMR proteins. In most cases, *MMR* pathogenic variants result in loss of protein expression in the tumor tissue. Nonetheless, there are cases where pathogenic variants negatively alter protein function but do not alter its immunogenicity, due to protein heterodimerization [41,42].

In our cohort, we demonstrated a high prevalence of *MMR* pathogenic variants in a cohort of patients who also presented *TP53* mutations. Nevertheless, we could not identify any difference in tumor behavior associated with the presence of variants in both genes. In LS patients with colorectal cancer, it was possible to identify that *TP53* variants modulated the age of development of CRC, with the age of tumor development being earlier in patients with *TP53* mutations than in LS patients who presented wild-type (WT) *TP53* [43]. In a breast cancer cohort, the association of *TP53* variants with *MSH* variants also impacted the age at diagnosis [44]. Young et al. studied the in vivo association between *MSH6* and *TP53* and their combined role in the suppression of tumorigenesis, cell survival, and genomic stability. They demonstrated that *TP53* and *MSH6* are functionally inter-related and that concomitant variants in both genes lead these tumor suppressors to act together to accelerate tumorigenesis [45].

Heath et al. reported no significant difference in the prevalence of childhood cancer in LS families, when compared to that of controls [46]. Although we showed positive data about the prevalence of *MMR* pathogenic variants in an ACT pediatric cohort, we cannot indicate or establish a recommendation for surveillance in this group. Further follow-up studies are necessary to prove whether *MMR* variant carriers who have already presented ACT are more likely to develop other cancers at an early age. Whether the association of *MMR* variants with *TP53* variants could affect tumor behavior, in turn impacting clinical outcomes and the survival rate, also needs to be addressed. However, we do encourage patients with both mutations to be followed with surveillance, according to the U.S. Multi-Society task force guideline [22] and the modified Toronto protocol [47].

## 4. Materials and Methods

Thirty-six pediatric patients with adrenocortical tumor were included in this cohort. All patients underwent adrenalectomy due to ACT. Patients had been followed in a unique tertiary center, the Faculty of Medicine of the University of São Paulo. All patients or their legal responses who donated adrenocortical tumor tissue and underwent genetic analyses gave written informed consent for tissue, clinical data, and blood sample collection. The study was approved by the ethics committee (16.686/2017) of the University of São Paulo. Detailed clinical data, including clinical and hormonal presentations, treatments, follow-up data, and survival data, in addition to histopathological data, were collected for all patients.

### 4.1. Immunohistochemistry

#### 4.1.1. Tissue Microarray (TMA) and Immunohistochemical Analysis of the TMA

Representative areas of 45 pediatric adrenocortical tumors from 45 different patients (viable tumor tissue without necrosis) were identified on hematoxylin and eosin-stained slides and marked on paraffin donor blocks. The spotted areas of donor blocks were punched (1.0 mm punch), and the cores were mounted into three paraffin blocks using a precision microarray instrument (Beecher Instruments, Sun Prairie, WI) with a total of 3 cores per patient/sample. One set of three slides was selected (one slide from each of the three TMA paraffin blocks of the triplicate) for staining with anti-MLH1, anti-PMS2, anti-MSH2, and anti-MSH6 ready-to-use Leica Biosystems antibodies (PA0610, PMS2-L-CE, PA0048, and PA0804, respectively). The reactions were carried out in an automated Leica Biosystems Bond-II machine (Leica Biosystems, Nussloch, Germany), which performed an automated modified immunoperoxidase immunohistochemical method with heat antigen retrieval.

TMA samples were included in the analysis only if at least one core with an internal nuclear positive control in either vessels or stromal cells was available after the staining procedure. Nuclear staining was evaluated as negative-deficient immunoexpression (no nuclear staining on neoplastic cells) or positive-preserved immunoexpression (nuclear staining present on neoplastic cells, regardless of percentage of positive cells) for the four antibodies.

#### 4.1.2. Formalin-Fixed and Paraffin-Embedded Tumor Sections

Formalin-fixed and paraffin-embedded tumor sections were cut at 4 μm thickness and stained on a Ventana Bench Mark Ultra Autostainer (Ventana Medical System, Tucson, Arizona) with the following antibodies: MLH-1 (clone M1, Ventana), MSH2 (clone G219-1129, Ventana), MSH6 (clone SP93, Ventana), and PMS2 (A16-4, Ventana). Rabbit monoclonal primary antibodies and paraffin-embedded tissue sections were utilized for these assays. The specific antibodies were visualized using an OptiView DAB IHC Detection Kit (Cat. No. 760-700), followed by an OptiView Amplification Kit (Cat. No. 760-099). Nuclear staining was also evaluated as negative-deficient immunoexpression (no nuclear staining on neoplastic cells) or positive-preserved immunoexpression (nuclear staining present on neoplastic cells, regardless of percentage of positive cells) for the four antibodies.

### 4.2. Genetic Analysis

#### 4.2.1. Panel Sequencing and Data Analysis

Genomic DNA was extracted from peripheral blood using the salting-out method from 36 patients. The genomic DNA was analyzed by the research assay HNPCC MASTR Plus (Agilent Technologies, Santa Clara, CA) to identify single-nucleotide variants in four genes associated with Lynch syndrome (*MLH1, MSH2, MSH6,* and *PMS2*) and the 3′ UTR of EPCAM, according to the manufacturer’s instructions. The sequences were generated using the MiSeq platform (Illumina, San Diego, CA) running in paired-end mode. Reads were aligned to the GRCh37/hg19 assembly of the human genome with the Burrows–Wheeler (BWA-mem) aligner. Variant calling included single-nucleotide variants, small insertions, and deletions, and was performed using the Freebayes platform. The resulting data (in variant call format) were annotated with ANNOVAR. The median coverage of the target bases was 7139.12X, with 100% of the targets having bases with ≥10× coverage.

The variants were screened for rare variants (minor allele frequency <0.1% in public and in-house databases) located in exonic regions and consensus splice site sequences. Subsequently, variant filtration prioritized their potential to be pathogenic: loss-of-function variants and variants predicted to be pathogenic by multiple in silico programs. The sequencing reads carrying candidate variants were inspected visually using the Integrative Genomics Viewer (Broad Institute, Cambridge, MA).

The databases gnomAD exome data, gnomAD genome data, and CllinVar, and computational prediction sites DANN, DEOGEN2, EIGEN, FATHMM-MKL, M-CAP, MVP, MutationAssessor, MutationTaster, PrimateAI, REVEL, and SIFT were considered for classification of the variants. Additionally, the International Society for Gastrointestinal Hereditary Tumours (InSiGHT) variant database was also used, in order to characterize variants according to MMR classification (Classes 1–5). 

Copy number variation analyses were performed using Copy Number Targeted Resequencing Analysis (CONTRA), a software which is able to call CNVs for a target region based on the normalized depth of coverage [48]. The variants were classified, according to the ACMG (American College Medical Genome) guidelines, by the Varsome platform [49].

#### 4.2.2. Sanger Sequencing

Sanger sequencing was used to confirm the allelic variants found and for the segregation analysis. The region corresponding to the allelic variants in the *PMS2*, *MLH1,* and *MSH6* genes (GenBank accession numbers NM_000535.7, NM_001354630, and NM_000179, respectively) was amplified by polymerase chain reaction (PCR), followed by automatic sequencing of the products using the Sanger method. The PCR primers and conditions are available upon request.

Sanger sequencing was also used to confirm the allelic variants of the *TP53* gene.

#### 4.2.3. MLPA

MLPA analysis was carried out using the commercial kits P008-C1 PMS2 and P003-D1 MLH1/MSH2 (MRC Holland), which are designed to detect exonic deletions/duplications across the coding regions of *PMS2*, *MLH1,* and *MSH2*. The assays were performed according to the manufacturer’s instructions. In brief, genomic DNA (20 ng) was denatured and hybridized overnight with the MLPA probe mix. After ligation, PCR amplification was carried out with specific SALSA FAM PCR primers. Amplicons were run on an ABI3130XL genetic analyzer (Applied Biosystems) using the ROX size standard. The peak area for each fragment was measured with GeneMapper 5.0 (Applied Biosystems). The obtained dates were analyzed by Coffalyser.Net (MRC Holland).

## 5. Conclusions

*MMR* genes may play a role in pediatric adrenocortical tumorigenesis, as seen in adult cohorts. Whether the association with *TP53* variants can influence tumor behavior and promote early clinical presentation remains unclear. However, patients with both mutations must be followed, according to the U.S. Multi-Society Task Force guidelines and the modified Toronto protocol, and genetic counselling must be offered to the proband family.

## Figures and Tables

**Figure 1 cancers-12-00621-f001:**
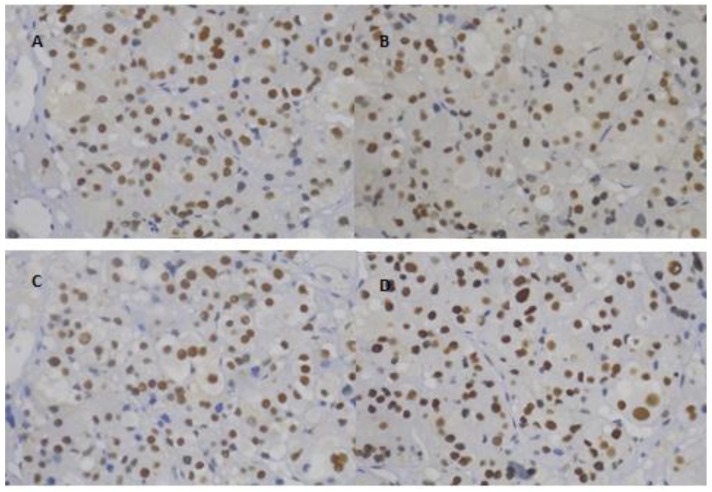
Representative adrenocortical tumor of a female patient, 2.7 years of age, with mutation on *TP53* (c.818G > A, p.Arg273His) and presenting wild-type *MMR* genes. These sections present preserved MLH1 (**A**), PMS2 (**B**), MSH2 (**C**), and MSH6 (**D**) immunostaining (positive nuclear staining), 400×.

**Figure 2 cancers-12-00621-f002:**
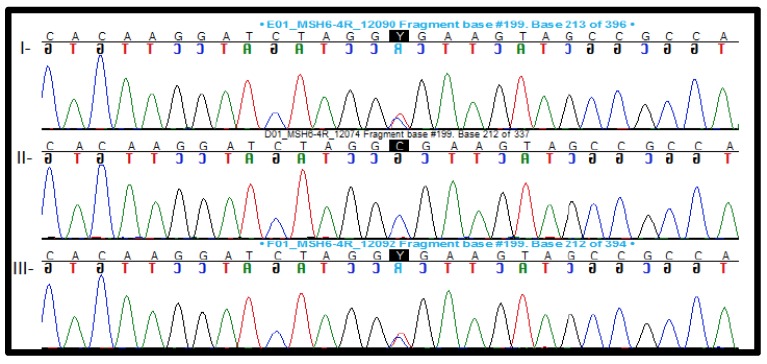
Figure of Sanger sequencing. I—Index case: patient #33 presenting a nonsense variant in the *MSH6* gene (c.328C > T, p.Arg110*; rs63750019); II—mother of index case; and III—father of index case.

**Table 1 cancers-12-00621-t001:** Next-generation sequencing (NGS) analysis.

Patients	Gender	Age (yrs)	Clinical Presentation	Metastasis	Global Survival Time (yrs)	*MMR* Genes Variants	InSiGHT DNA Variant Database	ACMG Classification	
**#6**	M	2.2	V	No	7.54	*MLH1*-c.1500_1502del p.Ile501del	Class 3: uncertain	Likely pathogenic	PM1; PM2;PM4; PP3
**#9**	M	2.1	V	No	8.19	*MLH1*-c.1808C > T p.Pro603Leu	Not identified	Likely pathogenic	PM1, PM2, PP2, PP3
**#16**	F	3.3	V/C	Yes	8.07	*MSH6*-c.2420A > G p.Tyr807Cys	Not identified	VUS	PM2, PP3, BP1
**#21**	F	3	V/C	No	31.61	*MSH6*-c.359T > C p.Ile120Thr	Not identified	Likely benign	PM1, PM2, BP3, BP4
**#33**	M	11.5	V/C	No	3.14	*MSH6*-c.328C > T p.Arg110X	Not identified	Pathogenic	PVS1, PM1, PM2, PP3, PP5
**#35**	M	2.3	V	No	0.08	*MSH6*-c.643G > T p.Val215Leu	Not identified	Likely benign	PM1, PM2, BP3, BP4

All the patients presented *TP53* Arg337His allelic variant. M: Male; F: Female; V: Virilization; **C**: Cushing’s syndrome; VUS: Variant of uncertain significance. PVS1—Null variant (nonsense, frameshift, canonical ±1 or 2 splice sites, initiation codon, single or multiexon deletion) in a gene where loss of function (LOF) is a known mechanism of disease. PM1—Located in a mutational hot spot and/or critical and well-established functional domain (e.g., active site of an enzyme) without benign variation. PM2—Absent from controls (or at extremely low frequency, if recessive) in Exome Sequencing Project, 1000 Genomes Project, or Exome Aggregation Consortium. PM4—Protein length changes as a result of in-frame deletions/insertions in a non-repeat region or stop-loss variants. PP2—Missense variant in a gene that has a low rate of benign missense variation and in which missense variants are a common mechanism of disease. PP3—Multiple lines of computational evidence support a deleterious effect on the gene or gene product (conservation, evolutionary, splicing impact, etc.). PP5—Reputable source recently reports variant as pathogenic, but the evidence is not available to the laboratory to perform an independent evaluation. PB1—Missense variant in a gene for which primarily truncating variants are known to cause disease. PB2—Observed in trans with a pathogenic variant for a fully penetrant dominant gene/disorder or observed in cis with a pathogenic variant in any inheritance pattern. PB3—In-frame deletions/insertions in a repetitive region without a known function. PB4—Multiple lines of computational evidence suggest no impact on gene or gene product (conservation, evolutionary, splicing impact, etc.).

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
