# Peer review of "High Prevalence of Alterations in DNA Mismatch Repair Genes of Lynch Syndrome in Pediatric Patients with Adrenocortical Tumors Carrying a Germline Mutation on TP53"

_cancers, 2020, doi:10.3390/cancers12030621_

Round 1
Reviewer 1 Report
The manuscript of Brondani et al. is interesting. With regard to the results obtained, the discussions are well argued. I would like to make only small notes on the interpretation of the variants identified in the MMR genes. It is good that the authors refer to INSIGHT as the mutation database. Therefore, the authors should reformulate the interpretation of the variants taking into account this database. Furthermore, the caption of table 1 should be better specified and the table modified by replacing the ACMG classification with the Insight classification. Finally, there is no hint in the description of the cases regarding the family history of these young patients ...
Reviewer 2 Report
This manuscript by Brondani et al. describes the high prevalence of variants in Lynch syndrome-related genes in pediatric adrenocortical tumor patients with a TP53 germline mutation. Considering rarity of the disease, this manuscript potentially provides important information. However, there are several concerns need to be addressed.
Major concerns:
- One of the major findings of this study is the identification of likely pathogenic MMR gene variants in a subset of pediatric patients with adrenocortical tumors with a TP53 variant. However, the adrenal tumor tissue showed a positive expression of MMR gene products which is unusual in Lynch syndrome. Alternative technique such as microsatellite instability (MSI) testing should be considered to assess pathologic significance of the identified MMR gene variants.
- Of the 35 patients with successful NGS analysis, six patients presented MMR gene variants. In these six patients with MMR variants, was there any family history of Lynch syndrome-associated tumors? This information should be included in the manuscript.
- Study design: It is unclear how the patients were selected. In the section 2.1. “Patients”, the authors state that “36 patients were included in this retrospective study”. On the other hand, in the section 4.1. “Immunohistochemistry”, there is a description of “45 pediatric adrenocortical tumors from 45 different patients”. Please clarify this.
- The description of immunohistochemistry (IHC) is a bit confusing. In this study, two different IHC studies were performed, one with tissue microarray (TMA) and the other with formalin-fixed paraffin embedded tumor sections. The staining protocols and primary antibodies appear to be different between these two. The authors should provide explanation for the purpose of these two IHC studies. It is also unclear the data in the results section represent the TMA study or FFPE tumor study. The IHC results should be better presented.
Minor comments:
- The number of samples for IHC that appears in the abstract (n=26) does not match with one described in the main text (n=36). Please double check.
- It would improve the manuscript if the author could provide a (supplemental) figure of Sanger sequencing results for the confirmation of NGS identified MMR gene variants.
- Figure 1: Please provide additional information of the sample, including genetic background, in the figure legend.
- For patients #6 and #33, please provide which computational analysis programs were used. If the same programs were used in all cases, the information can be described in the methods section.
- Immunohistochemistry: How sure the authors that the primary antibodies used in this study are specific? Was any of them validated in the previous studies?
- Genetic analysis: Please describe the method for genomic DNA isolation.
Round 2
Reviewer 2 Report
The authors appropriately answered to most of the reviewer comments. The manuscript improved significantly. However, the supplemental figure of Sanger sequencing results for the confirmation of NGS-identified MMR gene variants was not included in the revised manuscript. Please provide the figure.
Author Response
Dear reviewer
The supplemental figure of Sanger sequencing results for the confirmation of NGS-identified MMR gene variants was included in the revised manuscript (Line419-420), and all the Sanger sequencing was attached in the supplemental figure (Line 308).
The manuscript has undergone English language editing by MDPI.
Thank you.
